# Analysis of the Effectiveness of Coordinated Care in the Management of Pharmacotherapy of Patients with Hypertension and Comorbidities in Primary Care—Preliminary Reports

**DOI:** 10.3390/healthcare12111146

**Published:** 2024-06-05

**Authors:** Aleksandra Galic, Anna Tyranska-Fobke, Aleksandra Kuich, Andrzej Zapasnik, Marlena Robakowska

**Affiliations:** 1Faculty of Health Sciences with the Institute of Maritime and Tropical Medicine, Medical University of Gdańsk, 80-210 Gdańsk, Poland; 2Department of Public Health & Social Medicine, Medical University of Gdańsk, 80-210 Gdańsk, Poland; 3BaltiMed Gdansk Clinic, 80-041 Gdańsk, Poland

**Keywords:** coordinated care, hypertension, primary care, pharmacotherapy

## Abstract

Hypertension (HTN) is the dominant cause of cardiovascular disease and premature death worldwide. Also in Poland, the number of people with HTN is steadily increasing. In order to improve care for patients with HTN and other chronic diseases, a pilot of the POZ PLUS coordinated-care model was introduced. The pilot ran from 1 July 2018 to 30 September 2021 at 47 facilities nationwide. The purpose of this study was to conduct a preliminary analysis of the effectiveness of this model of care. The study focused on the management of pharmacotherapy in patients with hypertension and other comorbidities. The study included a group of 90 patients with HTN. Fifty-nine people were in the coordinated-care study group and 31 in the control group. Data were collected from electronic medical records. The analysis showed a trend toward greater blood-pressure reduction in patients under coordinated care (−4 mmHg difference in systolic blood pressure between the second and first visits and −2 mmHg difference in diastolic pressure between the second and first visits, *p* = 0.180 and *p* = 0.156). This suggests the preliminary conclusion that coordinated care in the PCP plus model might have positively affected the outcomes of patients with HTN. Further studies on the subject are planned.

## 1. Introduction

Hypertension (HTN) is the leading cause of cardiovascular disease and premature death worldwide [1]. HTN is responsible for more than 10 million preventable deaths globally each year [2]. In 2019, the global age-standardized prevalence of hypertension in adults aged 30–79 years was 32% in women and 34% in men [3]. In Poland in 2020, according to data from the National Health Fund (NFZ), the number of hypertensive patients was 9.94 million adults. Most patients are aged 55–74 years [4]. The prevalence of HNT among both sexes increases with age [5,6].

In view of the high prevalence of chronic diseases, including HNT, in the Polish population, the pilot of the POZ PLUS model ran from 1 July 2018 to 30 September 2021. The main goal of the pilot was to test the assumptions of the coordinated-care model in primary care (POZ) for patients with chronic diseases. Patient participation in the pilot was entirely voluntary and was preceded by an appropriate written statement. As part of the care, once included in the disease management program, the patient had the support of a physician and primary care nurse, a specialist physician, a nutritionist, a physiotherapist, a psychologist, a coordinator, and a health educator. As part of the pilot project, the primary care physician was able to quickly contact a specialist and consult a given case. An important role was played by a coordinator who took care of the exchange of information between medical personnel, the patient, and external entities, as well as supporting and organizing the treatment process. Coordination of the entire care process involved, among other things, scheduling appointments and examinations well in advance, reminding the patient of appointments via email or SMS, and actively monitoring the patient’s implementation of individual treatment recommendations.

A patient could be enrolled in the DMP disease management program on the basis of an adult balance examination or documented medical history. The program included patients who were eighteen years of age or older and had a diagnosis of at least one chronic disease from a list of eleven diseases. These included type II diabetes mellitus, spontaneous hypertension, chronic coronary artery disease, chronic heart failure, persistent atrial fibrillation, bronchial asthma, COPD, hypothyroidism, parenchymal and nodular goiter of the thyroid gland, peripheral osteoarthritis, and spinal pain syndromes. The patients who qualified to participate in the DMP program were included in a regimen that included, among other things, comprehensive advice from a primary care physician, follow-up advice from a primary care physician, consultation with a specialist, and educational or dietary advice.

Comprehensive counseling could take place up to three times a year, depending on the number of diseases diagnosed in the patient from different groups of disciplines. As part of such a visit, the doctor worked out an individual medical care plan (IPOM) together with the patient, taking into account the patient’s condition, after diagnostic tests. The IPOM included information regarding the patient’s planned diagnostic check-ups, specialized consultations, pharmacotherapy, or self-monitoring. Subsequent comprehensive counseling was used for in-depth monitoring of the patient’s condition and possible modification of the IPOM. During the comprehensive counseling, the doctor was able to consult the patient’s condition with other specialists. A follow-up visit, additionally funded under the pilot, could take place one to three times over the next 12 months. Scheduled in advance, it was intended, among other things, to check the implementation of IPOM, especially in terms of achieving the therapeutic goal or the doctor’s recommendations, including pharmacotherapy. Consultations of domain specialists (doctor–patient or doctor–doctor), were ordered by the outpatient doctor individually, depending on the patient’s defined needs. Educational counseling was implemented after it was recommended in the IPOM. The main task of educational counseling was to strengthen the patient’s competence for self-management and to actively involve the patient in the treatment process. Educational counseling was carried out by a nurse or health educator. It could take place individually or in a group of two to six patients with coinciding diseases. One cycle of 12 months was provided; the cycle involved three counseling sessions, with an interval of no more than 4 months between the first and third visits.

The doctor under IPOM could order dietary counseling for the patient. This was a cycle of three counseling sessions; one cycle could take place once every 12 months, depending on the patient’s needs. The first visit consisted of pointing out the patient’s mistakes, learning how to eliminate them, and developing an intervention plan. The second was to analyze what the patient managed to implement, and the third was to evaluate the effectiveness of the intervention.

A patient’s participation in the DMP program could be terminated if the patient’s condition did not allow him or her to be treated by the PCP, the patient himself or herself opted out of the pilot, or the patient did not follow the recommendations from the IPOM [7]. Under the pilot, the patients had access to an expanded range of diagnostic tests so that they did not have to be referred to a specialist physician’s clinic for them. For a person diagnosed with hypertension under the DMP, the PCP could order cardiac stress testing, cardiac echo, RR Holter, and microalbuminuria testing, among others [8].

The pilot study was conducted at 47 selected facilities nationwide [9]. During it, 37,000 patients signed consents to participate in the disease management program. A total of 508,666 services were provided to those covered by the DMP [10].

On the basis of the pilot, a coordinated-care model was introduced into primary care in Poland from 1 October 2022 [11]. In the period between the end of the pilot (1 October 2021) and the introduction of coordinated care in primary care, care based on the assumptions of the pilot continued at the facility where the study was conducted. Patients continued to have access to educational and dietary advice, as well as a broader package of diagnostic tests.

The purpose of the presented study is a preliminary analysis of the effectiveness of coordinated care in the management of pharmacotherapy of patients with hypertension and comorbidities. The main objective of the analysis is to evaluate the impact of coordinated care for HTN patients under the POZ PLUS model on changes in systolic and diastolic blood-pressure values and changes in the pharmacotherapy of patients under coordinated care compared to patients in the standard model of care in primary care.

## 2. Materials and Methods

The present study selected a sample of 90 patients with hypertension (essential (primary) hypertension I10) receiving coordinated care under the POZ PLUS pilot at the BaltiMed Gdansk primary care facility located in the Pomeranian Voivodeship in Northern Poland. This facility is one of 47 where the pilot of POZ plus was conducted. All patients and their data come from this one facility. The facility has more than 10,000 patients under its care in primary care. In 2023, it provided 43,393 consultations in primary care and 21,061 in outpatient specialist care.

The patient sample consisted of patients belonging to the study group (with the intervention in the form of coordinated care) and the control group (without the intervention), with the inclusion criterion being data from two consecutive visits, the first of which took place after the implementation of this model of care, i.e., after 1 July 2018, and the last of which took place by 31 December 2022. Limiting the number of visits to two was due to the insufficient data available for the control group. The selection of patients for the study consisted of analyzing the data of the study clinic for the presence of patients meeting the following inclusion criteria: being under the care of a primary care physician at the Baltimed clinic since 1 January 2017, being under coordinated care between 1 July 2018 and 31 December 2022, a diagnosis of I10, and age between 40 and 75 years. Patients under the care of a primary care physician in the Baltimed clinic from 1 January 2017, not subject to coordinated care in the period from 1 July 2018 to 31 December 2022, with a diagnosis of I10, and an age between 40 and 75 years were included in the control group. The data came from the patients’ electronic medical records. The study did not require additional patient consent for data use. The data were anonymized. The study collected the following data: weight, height, smoking status, blood pressure, glucose, total cholesterol, LDL, TG, HDL, non-HDL, potassium, creatinine, eGFR, uric acid, and urinary protein, among others. Information on blood-pressure measurements came either directly from the visit, when the doctor measured the patient’s blood pressure, or from the patient’s home measurements, when he or she reported with a blood-pressure diary for the visit. In the case of telephone visits, the doctor could ask the patient to measure his or her blood pressure at the time of the visit and then record it in the chart.

In this statistical analysis, the conventional significance level of α = 0.05 was adopted. To assess the normality of the distributions of the variables, the Shapiro–Wilk test was used. For the analysis of differences between two independent groups, in cases where the distributions did not meet the assumption of normality, the non-parametric Wilcoxon rank sum test (another name is the Mann–Whitney test) was used. For categorical variables, Pearson’s chi-square test was used and, in situations requiring it due to small sample sizes, Fisher’s exact test. Analyses were performed using the statistical language R (version 4.3.1; R Core Team, 2023) [12] on Windows 10 Pro 64 bit (version 19045), using the packages report (version 0.5.7) [13], ggstatsplot (version 0.12.1) [14], gtsummary (version 1.7.2) [15], readxl (version 1.4.3) [16], and dplyr (version 1.1.3) [17].

The study received approval from the Independent Bioethics Committee at the Medical University of Gdansk, registered under the number KB/612/2023.

## 3. Results

A sample of 90 patients was analyzed, of which, 59 belonged to the study group and 31 to the control group. Detailed characteristics of the sample are shown in Table 1.

The data presented shows an almost even gender distribution in the study population, with a slight advantage for women The proportions are similar in both groups (study and control), indicating a well-balanced gender distribution in terms of potential impact on the study results. The mean age in the study population was 72 years, with slight differences between the median age in the study and control groups. The presence of comorbidities, such as diabetes and hypercholesterolemia, was also reported.

Table 2 presents data on the hypertension drug therapies used, along with the results of blood-pressure measurements obtained at the first visit. Single-pill combinations (SPC) referred to in the table are drugs containing at least two API-active pharmaceutical ingredients dedicated to the treatment of hypertension [18]. The 2013 and 2018 ESH/ESC guidelines favored the use of two antihypertensive drugs as an SPC because reducing the number of pills to be taken daily improves adherence to treatment and increases the rate of BP control [19].

In the overall sample (N = 90), the most commonly used drugs were ACEs, which were used by 50% of patients. Other drug groups, such as beta-blockers, calcium channel blockers, thiazide diuretics, or thiazide-like diuretics, also showed no significant differences between groups. A small difference was observed in the use of other drug groups (*p* = 0.090), which may suggest a trend, although it did not reach the conventional level of statistical significance (*p* < 0.05).

The median systolic blood pressure in the entire sample was 148 mmHg. The median was higher in the study group than in the control group. For diastolic pressure, a median of 80 mmHg was obtained for the entire sample. However, a lower median was observed in the study group compared to the control group. As for the hemodynamic analysis, there were no statistically significant differences in systolic or diastolic blood-pressure values between the study and control groups.

Table 3 presents data on the hypertension drug therapies used, together with the results of the blood-pressure measurements obtained at the second visit.

Beta-blockers were used by 28.89% of the total sample, with a higher percentage in the study group than in the control group. A similar percentage of patients in the total group were taking calcium channel blockers, with a slightly higher percentage in the study group. Thiazide/thiazide-like diuretics were used by 25.56% of the total sample, with a slightly lower percentage in the study group compared to the control group. Loop diuretics were used by 14.44% of the total sample, with a slight difference between the study group and the control group. The number of patients using aldosterone antagonists represented a small percentage of the total sample (3.33%). ACEs were the most commonly used drug group in the entire sample, with similar percentages in the study group and the control group. ARBs were used by 10.00% of the sample overall. Other groups of antihypertensive drugs were used by 7.78% of the total sample. SPCs were used by 24.44% of the total sample. Analysis of the data showed no statistically significant differences in the use of the different groups of antihypertensive drugs between the trial and control groups.

The median systolic blood pressure across the sample was 142 mmHg, while the median diastolic pressure was 80 mmHg. The *p*-values for both blood-pressure parameters (systolic and diastolic pressure) are significantly higher than the accepted threshold for statistical significance, indicating that there were no significant differences between the study and control groups in terms of median blood-pressure levels at the second follow-up visit.

Table 4 shows the analysis of the time distance between the first and second follow-up visits, the changes in blood-pressure results, and the number of patients who achieved at least a blood-pressure control of <140/90 mmHg in the overall sample and stratified by the patient group.

The median time between visits for the entire sample was 1.84 months, with an interquartile range of 0.92 to 3.91 months. The median was higher for the study group compared to the control group. Analysis of the difference in systolic blood pressure between the second and first visits showed a median of −3.00 mmHg for the sample overall, with IQRs ranging from −10.00 to 9.00 mmHg. In the study group, the median was −4.00 mmHg, and in the control group, it was 0.00 mmHg.

On the basis of the extended analysis of the data, significant differences were observed in the dynamics of systolic pressure changes between the sexes. Men showed a significantly greater decrease in systolic pressure, with a median of −9.00 mmHg, which contrasts with the female group, where the median change in pressure showed no change and was 0.00 mmHg. The results are presented in Figure 1.

For the difference in diastolic pressure between visits, the median for the entire sample was −4.00 mmHg, with an IQR of −10.00 to 5.00 mmHg. In the study group, the median change was −2.00 mmHg, while in the control group, the median change was −7.00 mmHg.

At the first visit, a total of 29 patients in both groups achieved at least a blood-pressure control of <140/90 mmHg. Of those, 16 were in the intervention group, and 13 were in the control group. In contrast, at the second visit, a total of 36 patients in both groups achieved at least a blood-pressure control of <140/90 mmHg. Of those, 25 were in the intervention group, and 11 were in the control group. As part of assessing the dynamics of antihypertensive drug use, it is crucial to understand patients’ dosing adaptations during the interval. Four different behavioral patterns of drug use were examined: continuous non-use (the patient did not take the drug during either the first or second visit), inclusion of drug use (the patient who did not use the drug during the first visit began taking it by the time of the second visit), continuous use (the patient continued taking the drug between visits), and exclusion of drug use (the patient used the drug during the first visit and did not use it during the second visit).

The data collected paint a picture of changes in antihypertensive drug dosage both in the entire study sample and by patient group (Table 5).

An analysis of data on the modification of used beta-blocker medications among patients treated for hypertension showed varying trends in the management of drug treatment between visits. In the total sample, the largest proportion of patients maintained a state of non-adherence to this class of medications. There were also cases of exclusion from the use of this class of drugs, which accounted for 12.22% of the entire study group. In the study group, this accounted for 8.47%, while in the control group, the rate was much higher at 19.35%, suggesting a potential difference in pharmacotherapy between the groups.

In the analyzed dataset on the modification of the use of calcium channel blocker drugs among hypertensive patients, we observe a distribution that shows no statistically significant differences in the whole group and within the study and control groups. In the total sample, the majority of patients did not use medications in this category. Ongoing use of this group of drugs, an indicator of continued pharmacotherapy, was observed in 15.56% of the patients overall. In the study group, the use of drugs of this group was maintained by 18.64% of patients, which exceeds the rate for the control group. Analysis of the frequency of inclusion of medications from this group showed that 13.33% of the total patients started using the drug between visits. In the study and control groups, these percentages were similar, 13.56% and 12.90%, respectively.

Of the total number of patients, the majority were not treated with thiazide/thiazide-like diuretics. The inclusion of drug use between visits was recorded in 11.11% of patients.

The analysis of dosage changes of antihypertensive drugs from the loop diuretic group indicates a generally low dynamics of pharmacotherapy modification among patients of both compared groups. Among the patients analyzed, the predominant therapeutic regimen is no use of loop diuretics, as observed in 84.44% (n = 76) of the entire sample. Discontinuation of loop diuretics between visits is an infrequent event and affected only 1.11% of patients, only in the control group.

The proportion of patients who were not treated with ACE drugs was 38.89%, with a higher proportion in the study group compared to the control group. A portion of the group discontinued taking ACE between visits, with a similar trend in both groups. Reasons for discontinuation could be multiple, including the occurrence of side effects, such as ACE-induced cough or insufficient therapeutic response, motivating a change to drugs from another therapeutic group. Maintaining a steady dose and steady ACE intake was reported in 36.67% of patients, reflecting stability in the pharmacotherapy of both groups.

ARBs are used in a small number of patients to treat hypertension in the study group.

An analysis of data on the use of alpha-blockers among patients treated for hypertension showed that these drugs were not used in any case in the entire study group.

Antihypertensive drugs of other groups were rarely used in both groups of patients, and changes in pharmacotherapy showed no statistically significant differences between the study and control groups.

SPCs were not a dominant part of the pharmacotherapy regimen in the analyzed group of hypertensive patients. In both study groups, similar pharmacotherapeutic regimens were noted for this group of drugs.

## 4. Discussion

Based on the analysis presented, it can be concluded that patients in both the study group and the control group showed no statistically significant differences in the time distance between visits, indicating a comparable level of adherence to the recommendations for regularity of visits.

In terms of blood-pressure control, despite the lack of statistically significant differences, a trend can be observed that suggests a greater reduction in systolic and diastolic blood pressure in the study group, which may indicate a positive effect for the intervention used. In clinical practice, blood-pressure control is a very important element in the strategy to prevent complications of hypertension, such as coronary heart disease, heart failure, cerebrovascular disease, lower extremity arterial disease, chronic kidney disease, and atrial fibrillation [20].

Gender variations in blood-pressure response to treatment suggest that men may benefit more from therapeutic interventions aimed at lowering blood pressure, which may have significant implications for personalizing hypertension treatment strategies.

Further studies with larger numbers of participants and the use of more rigorous statistical methods are needed to confirm these observations. The *p*-value ranging from 0.156 to 0.457 for differences in systolic and diastolic blood pressure, while not indicative of statistical significance, does not allow for a clear rejection of the hypothesis that the intervention had no effect on blood-pressure changes. Therefore, these results can be a starting point for further, more detailed, analyses.

The analysis also showed no statistically significant differences in the use of specific classes of drugs between the study and control groups, which may suggest similar standards of hypertension treatment in the two groups. The lack of significant differences also indicates that the drug classes themselves were not a decisive factor in differences in blood-pressure control or clinical outcomes between the groups. Nevertheless, there are some trends in the pharmacotherapy provided, which may require further investigation in larger groups of patients.

Analyzing changes in adherence to individual medication groups provides key clues to adherence to treatment, the effectiveness of pharmacotherapy, and the identification of the need to modify treatment regimens in the context of comprehensive hypertension management. Patterns of drug use may reflect individual patient responses to therapy, the occurrence of adverse effects, changes in clinical recommendations, or fluctuations in drug availability. Tracking these changes is essential to ensure effective hypertension control and to minimize the risk of cardiovascular complications, which are often associated with suboptimal hypertension treatment. Thus, the analysis makes it possible to identify trends and differences in the management of hypertension pharmacotherapy, which is essential for assessing the effectiveness of adopted treatment strategies and understanding patients’ patterns of behavior between scheduled follow-up visits.

One of the key limitations of the study is the relatively small sample size. We conducted the study on a group of 90 participants, and there are plans for further analyses conducted on a larger number of participants. Another limitation may be that the data came from medical visits, and if something was not recorded correctly, it may not have been included in the analysis. Also, sometimes data were not available at all.

The choice of the appropriate drug group is crucial in the treatment of hypertension. According to the 2018 recommendations of the Polish Society of Family Medicine, the first-line drugs for the treatment of HTN are preparations of five basic groups of hypotensive drugs, according to the American and European guidelines for the treatment of NT. The basic drug groups include thiazide-like/thiazide diuretics, beta blockers, calcium channel blockers, ACEI, and ARBs [21]. The choice of one over the other is guided by patients’ comorbidities, such as chronic kidney disease, diabetes, heart failure, albuminuria, and race [22]. A common problem among patients with chronic diseases is poor adherence to prescribed medication [23]. Other groups of medications were also considered in the analysis, as all patients with hypertension were studied, not just those with a recent diagnosis. Patients with hypertension who take more pills have lower adherence rates. A single-pill combination can improve patients’ adherence [24]. The most commonly used drugs were angiotensin-converting enzyme inhibitors ACE. These drugs are sauced in the elderly for organ damage, high cardiovascular risk, and with comorbidities such as ischemic heart disease, kidney disease, or diabetes [21]. Trends in the frequency of prescribing particular groups of drugs as a first-line treatment for HTN have varied over time. In the UK, angiotensin-converting enzyme inhibitors were the most commonly prescribed drugs in primary care in 2018. In second and third place were calcium channel inhibitors and beta blockers, respectively [25]. In the present study, the list of the most commonly used drugs looked similar. There were slight differences at the first and second visits. There were cases of discontinuation of certain drug groups in pharmacotherapy and the inclusion of drugs between visits.

Hypertension is a global health issue among the adult population, with high morbidity and mortality rates [26]. For years, the effectiveness of various coordinated-care solutions has been studied in order to make HTN patient care as effective and comprehensive as possible. Various models of coordinated care for patients have been studied in different countries [27]. In the US, the Physician–Pharmacist Collaborative Model (PPCM) was studied. The intervention involved the pharmacist reviewing the patient’s medication history and, then, assessing the patient’s knowledge of medications, dosages, and blood-pressure control. Then, the pharmacist created a plan for the physician with recommendations to adjust therapy. A phone call up to 2 weeks after the study, and then, visits at 1, 2, 4, 6, and 8 months were also recommended. It was shown that patients achieved better outcomes than in standard care [28]. In Italy, a nurse-based model was studied in 2014. The intervention involved regular patient follow-up and weekly email alerts and phone calls to a nurse for 6 months. The emails included notifications regarding the need to adhere to a healthy lifestyle and recommendations in accordance with current physician orders, including pharmacotherapy [29]. The intervention group showed significant improvements in BMI and systolic and diastolic blood pressure. A similar nurse-based model was studied for 12 months in Brazil. The intervention began in February 2017 and consisted of nurse consultations, telephone contact, home visits, health education, and appropriate patient referrals. Intervention patients had significantly better outcomes in terms of blood-pressure drop, weight loss, and waist circumference. These patients had better adherence to treatment recommendations, including pharmacotherapy [30]. Medication adherence is critical for the effective management of hypertension; yet, half of patients with hypertension are non-adherent to medications [31]. It is important to educate patients about their disease, and it has been shown that strategies utilizing patient education and engagement are needed to improve medication adherence and blood-pressure control [32]. Through education, patients are more aware of their own disease and realize the importance and necessity of adhering to their doctor’s orders and taking prescribed medications regularly. A team approach is also important. Studies show the effectiveness of a team-based approach to blood-pressure management [33]. Adherence to medical recommendations for treatment and blood-pressure control is influenced by factors such as age gender, and education level. Studies have shown that being older, female, and of a lower educational level were more adherent to hypertension management [34]. Other factors that affect adherence include demographic, socioeconomic, and concomitant medical–behavioral conditions; therapy-related, healthcare team, and system-related factors and patient factors are associated with nonadherence [35]. Understanding the factors that contribute to patients’ noncompliance is important to be able to eliminate them and effectively treat and control hypertension.

Health systems should focus on care coordination, ensuring types of care per the healthcare needs at different stages of health conditions by a multidisciplinary team [36]. Common features of effective interventions included tailored communication with patients, the use of health information technology, and multidisciplinary collaboration [37]. Different solutions will work in different countries. International experience cannot be transferred in its entirety to Poland. Experience gained during program implementation is irreplaceable. It is also necessary to deeply understand the local context and conditions [38]. Coordinated care can significantly improve health care in Poland, but it is not the solution to all problems. Therefore, the implementation of coordinated care should be preceded by a well-documented pilot, followed by a gradual scale-up [39]. Patients with hypertension and other comorbidities have complex healthcare needs that are difficult to provide in primary care. However, there is strong evidence suggesting that patient-centered approaches to primary care are effective for managing complex multimorbidity [40]. It is, therefore, critical to tailor services and interventions in coordinated care to meet the health needs of patients so that they can be treated effectively in primary care.

Following and based on the pilot, the coordinated-care model in primary health care is being implemented in Poland from 1 October 2022. Implementation of this care is not mandatory for all primary care providers under contract with the National Health Fund. Facilities (NFZ) wishing to implement coordinated care may apply to the NFZ for an extension of the basic contract.

## 5. Conclusions

Based on the analysis, a trend toward greater blood-pressure reduction was observed in patients receiving coordinated care. This suggests a preliminary conclusion that coordinated care in the form of the analyzed model might have a positive impact on the outcomes of patients with HTN and comorbidities. At the same time, similar standards were observed in prescribing particular groups of drugs in both groups. This suggests that the standard of treatment in both groups was similar. However, further analysis of a larger group of patients, taking into account the influence of other factors on the preliminary results obtained, is needed to confirm these initial reports, which the authors of the article are currently working on.

## Figures and Tables

**Figure 1 healthcare-12-01146-f001:**
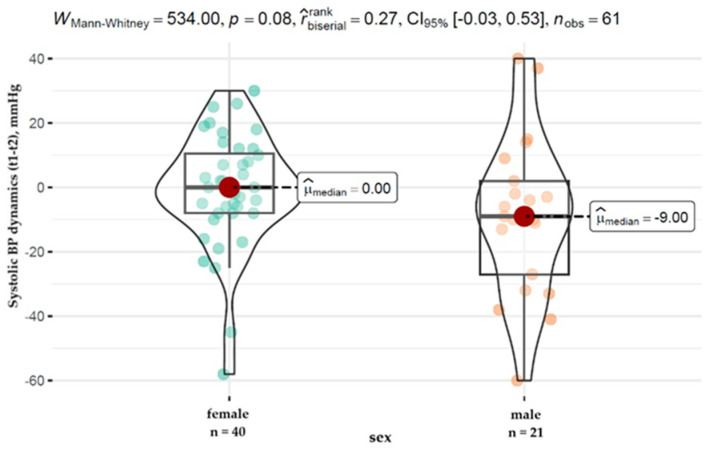
Distribution of changes in systolic pressure over time according to patient gender.

**Table 1 healthcare-12-01146-t001:** Characteristics of sociodemographic data along with characteristics of comorbidities at the first visit for the total sample and by group of patients.

Description	N	Total Sample N = 90 ^a^	Intervention Group n_1_ = 59 ^a^	Control Group n_2_ = 31 ^a^	*p* ^c^
sex	90				0.856
female		54.00 (60.00%)	35.00 (59.32%)	19.00 (61.29%)	
male		36.00 (40.00%)	24.00 (40.68%)	12.00 (38.71%)	
age on the day of visit, years	90	72.05 (69.80, 74.13) ^b^	72.01 (69.77, 73.46) ^b^	72.71 (70.67, 74.92) ^b^	0.203 ^d^
hypercholesterolemia	90	10.00 (11.11%)	8.00 (13.56%)	2.00 (6.45%)	0.484 ^e^
diabetes	90	15.00 (16.67%)	11.00 (18.64%)	4.00 (12.90%)	0.487 ^e^

^a^ n (%); ^b^ Mdn (Q1, Q3); ^c^ Pearson’s chi-squared test; ^d^ the Wilcoxon rank sum test; ^e^ Fisher’s exact test; Annotation: N—sample size, n—group size, Mdn—median, Q1—first quartile (25%), Q3—third quartile (75%), *p*—*p*-value of statistical test.

**Table 2 healthcare-12-01146-t002:** Characteristics of hypertension medications taken and blood-pressure results for the total sample and by patient group at the first visit.

Description	N	Total Sample N = 90 ^a^	Intervention Group n = 59 ^a^	Control Group n = 31 ^a^	*p* ^c^
Beta blockers	90	29.00 (32.22%)	20.00 (33.90%)	9.00 (29.03%)	0.639
Calcium channel blockers	90	24.00 (26.67%)	17.00 (28.81%)	7.00 (22.58%)	0.525
Thiazide and thiazide-like diuretics	90	22.00 (24.44%)	15.00 (25.42%)	7.00 (22.58%)	0.766
Loop diuretics	90	6.00 (6.67%)	3.00 (5.08%)	3.00 (9.68%)	0.411 ^e^
Aldosterone antagonist	90	1.00 (1.11%)	0.00 (0.00%)	1.00 (3.23%)	0.344 ^e^
Angiotensin-converting enzyme (ACE)	90	45.00 (50.00%)	28.00 (47.46%)	17.00 (54.84%)	
Angiotensin receptor blockers (ARBs)	90	8.00 (8.89%)	6.00 (10.17%)	2.00 (6.45%)	0.710
Other medication groups	90	6.00 (6.67%)	6.00 (10.17%)	0.00 (0.00%)	0.090 ^e^
Single-pill combinations (SPC)	90	20.00 (22.22%)	13.00 (22.03%)	7.00 (22.58%)	0.953
Systolic pressure, mmHg	70	148.00 (135.25, 166.75) ^b^	151.00 (137.00, 167.75) ^b^	140.00 (130.75, 157.00) ^b^	0.165 ^d^
Diastolic pressure mmHg	70	80.00 (76.25, 87.75) ^b^	80.00 (76.00, 85.00) ^b^	81.50 (79.00, 90.00) ^b^	0.146 ^d^

^a^ n (%); ^b^ Mdn (Q1, Q3); ^c^ Pearson’s chi-squared test; ^d^ the Wilcoxon rank sum test; ^e^ Fisher’s exact test. Annotation: N—sample size, n—group size, Mdn—median, Q1—first quartile (25%), Q3—third quartile (75%), *p*—*p*-value of statistical test.

**Table 3 healthcare-12-01146-t003:** Characteristics of hypertension medications taken and blood-pressure results for the total sample and by patient group at the second visit.

Description	N	Total Sample N = 90 ^a^	Intervention Group n = 59 ^a^	Control Group n= 31 ^a^	*p* ^c^
Beta blockers	90	26.00 (28.89%)	20.00 (33.90%)	6.00 (19.35%)	0.148
Calcium channel blockers	90	26.00 (28.89%)	19.00 (32.20%)	7.00 (22.58%)	0.339
Thiazide and thiazide-like diuretics	90	23.00 (25.56%)	13.00 (22.03%)	10.00 (32.26%)	0.291
Loop diuretics	90	13.00 (14.44%)	8.00 (13.56%)	5.00 (16.13%)	0.759
Aldosterone antagonist	90	3.00 (3.33%)	1.00 (1.69%)	2.00 (6.45%)	0.272 ^e^
Angiotensin-converting enzyme (ACE)	90	43.00 (47.78%)	27.00 (45.76%)	16.00 (51.61%)	0.598
Angiotensin receptor blockers (ARBs)	90	9.00 (10.00%)	7.00 (11.86%)	2.00 (6.45%)	0.713 ^e^
Other medication groups	90	7.00 (7.78%)	6.00 (10.17%)	1.00 (3.23%)	0.415 ^e^
Single-pill combinations (SPC)	90	22.00 (24.44%)	12.00 (20.34%)	10.00 (32.26%)	0.211
Systolic pressure, mmHg	76	142.00 (133.00, 153.50)	142.50 (132.25, 154.50)	140.50 (135.75, 149.00)	0.810
Diastolic pressure mmHg	75	80.00 (73.00, 88.00) ^b^	80.00 (73.00, 85.00) ^b^	78.00 (70.25, 90.00) ^b^	0.903 ^d^

^a^ n (%); ^b^ Mdn (Q1, Q3); ^c^ Pearson’s chi-squared test; ^d^ the Wilcoxon rank sum test; ^e^ Fisher’s exact test. Annotation: N—sample size, n—group size, Mdn—median, Q1—first quartile (25%), Q3—third quartile (75%), *p*—*p*-value of statistical test.

**Table 4 healthcare-12-01146-t004:** Characteristics of the time distance between visits, changes in blood-pressure results over time, and the number of patients who achieved at least blood-pressure control of <140/90 mmHg for the total sample and with stratification by patient group.

Description	N	Total Sample N = 90 ^a^	Intervention Group n = 59 ^a^	Control Group n = 31 ^a^	*p* ^b^
Time between first and second visits, months	90	1.84 (0.92, 3.91)	2.14 (1.03, 3.96)	1.41 (0.92, 3.70)	0.457
Difference in systolic blood pressure between the second and first visits (Δt2 − t1), mmHg	61	−3.00 (−10.00, 9.00)	−4 (−10.00, 3.75)	0.00 (−12.00, 16.00)	0.180
Difference in diastolic pressure between the second and first visits (Δt2 − t1), mmHg	61	−4.00 (−10.00, 5.00)	−2.00 (−7.50, 6.50)	−7.00 (−12.00, 2.00)	0.156
Number of patients who achieved at least blood-pressure control of <140/90 mmHg at the first visit	61	29	16	13	0.167
Number of patients who achieved at least blood-pressure control of <140/90 mmHg at the second visit	61	36	25	11	0.171

^a^ Mdn (Q1, Q3); ^b^ the Wilcoxon rank sum test; annotation: N—sample size, n—group size, Mdn—median, Q1—first quartile (25%), Q3—third quartile (75%), *p*—*p*-value of statistical test.

**Table 5 healthcare-12-01146-t005:** Changes in antihypertensive medications dosage between visits and comparison of differences between patient groups.

Description	N	Total Sample N = 90	Intervention Group N = 59	Control Group N = 31	*p*
medications from the beta-blocker group	90				0.189
non-use		53.00 (58.89%)	34.00 (57.63%)	19.00 (61.29%)	
exclusion of use		11.00 (12.22%)	5.00 (8.47%)	6.00 (19.35%)	
continuous use		18.00 (20.00%)	15.00 (25.42%)	3.00 (9.68%)	
inclusion of use		8.00 (8.89%)	5.00 (8.47%)	3.00 (9.68%)	
medications from the calcium channel blocker group	90				0.751
non-use		54.00 (60.00%)	34.00 (57.63%)	20.00 (64.52%)	
exclusion of use		10.00 (11.11%)	6.00 (10.17%)	4.00 (12.90%)	
continuous use		14.00 (15.56%)	11.00 (18.64%)	3.00 (9.68%)	
inclusion of use		12.00 (13.33%)	8.00 (13.56%)	4.00 (12.90%)	
medications from thiazide group	90				0.609
non-use		58.00 (64.44%)	39.00 (66.10%)	19.00 (61.29%)	
exclusion of use		9.00 (10.00%)	7.00 (11.86%)	2.00 (6.45%)	
continuous use		13.00 (14.44%)	8.00 (13.56%)	5.00 (16.13%)	
inclusion of use		10.00 (11.11%)	5.00 (8.47%)	5.00 (16.13%)	
medications from loop diuretic group	90				0.675
non-use		76.00 (84.44%)	51.00 (86.44%)	25.00 (80.65%)	
exclusion of use		1.00 (1.11%)	0.00 (0.00%)	1.00 (3.23%)	
continuous use		5.00 (5.56%)	3.00 (5.08%)	2.00 (6.45%)	
inclusion of use		8.00 (8.89%)	5.00 (8.47%)	3.00 (9.68%)	
medications from the aldosterone antagonist group	90				0.422
non-use		87.00 (96.67%)	58.00 (98.31%)	29.00 (93.55%)	
continuous use		1.00 (1.11%)	0.00 (0.00%)	1.00 (3.23%)	
continuous use		2.00 (2.22%)	1.00 (1.69%)	1.00 (3.23%)	
medications from the angiotensin-converting enzyme (ACE) inhibitor group	90				0.351
non-use		35.00 (38.89%)	26.00 (44.07%)	9.00 (29.03%)	
exclusion of use		12.00 (13.33%)	6.00 (10.17%)	6.00 (19.35%)	
continuous use		33.00 (36.67%)	22.00 (37.29%)	11.00 (35.48%)	
dose increase		10.00 (11.11%)	5.00 (8.47%)	5.00 (16.13%)	
medications from the angiotensin receptor blocker (ARB) group	90				0.906
non-use		80.00 (88.89%)	51.00 (86.44%)	29.00 (93.55%)	
exclusion of use		1.00 (1.11%)	1.00 (1.69%)	0.00 (0.00%)	
continuous use		7.00 (7.78%)	5.00 (8.47%)	2.00 (6.45%)	
inclusion of use		2.00 (2.22%)	2.00 (3.39%)	0.00 (0.00%)	
medications from the alpha-blocker group	90				n/d
non-use		90.00 (100.00%)	59.00 (100.00%)	31.00 (100.00%)	
medications from other groups	90				0.426
non-use		81.00 (90.00%)	51.00 (86.44%)	30.00 (96.77%)	
exclusion of use		2.00 (2.22%)	2.00 (3.39%)	0.00 (0.00%)	
continuous use		4.00 (4.44%)	4.00 (6.78%)	0.00 (0.00%)	
inclusion of use		3.00 (3.33%)	2.00 (3.39%)	1.00 (3.23%)	
single-pill combinations (SPC)	90				0.507
non-use		62.00 (68.89%)	42.00 (71.19%)	20.00 (64.52%)	
exclusion of use		6.00 (6.67%)	5.00 (8.47%)	1.00 (3.23%)	
continuous use		14.00 (15.56%)	8.00 (13.56%)	6.00 (19.35%)	
inclusion of use		8.00 (8.89%)	4.00 (6.78%)	4.00 (12.90%)	

## Data Availability

Data are contained within the article.

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
