# Peer review of "Analysis of the Effectiveness of Coordinated Care in the Management of Pharmacotherapy of Patients with Hypertension and Comorbidities in Primary Care—Preliminary Reports"

_healthcare, 2024, doi:10.3390/healthcare12111146_

Round 1

Reviewer 1 Report

Comments and Suggestions for Authors

1. Coordinated care is important and good to study this topic

2. The study should have randomized subjects- with adequate sample size in each arm to arrive at the effectiveness of the intervention

3. Current approach is not scientifically sound 

4. COst and other aspects to be covered to indicate if the coordinated approach is feasible in thousands of people who need this

Reviewer 2 Report

Comments and Suggestions for Authors

This study focused on managing medication in patients with hypertension and other comorbidities using a coordinated care model. There are several issues that should be considered.

1.      The conclusion of the study did not match the results obtained.

2.      The authors report that the analysis showed a trend toward greater blood pressure reduction in patients under coordinated care (-4 mmHg difference in systolic blood pressure between the 22 second and first visits and -2 mmHg difference in diastolic pressure between the second and first 23 visits, p 0,180 and p 0,156) This suggests the preliminary conclusion that coordinated care in the PCP 24 plus model positively affected the outcomes of patients with HTN. Does this minor shift, which may have been caused by other circumstances, lead to the study's conclusion?

3.      The authors include 47 facilities nationwide, however only 90 patients with HTN participated in the study. This means that around two patients from each facility participate. How does the author control the effect of demographics on the HTN control.

4.      The percent of female patients within group is higher than the male which could affect the obtained results. Do the authors make a regression analysis for controlling gender?

5.      What about the study’s inability to mask research assessors, and lack of information about participants’ adherence to recommendations for lifestyle and medication management of HTN. Generalizability of the model to other regions be limited.

6.       The results of this study showed small and non-significant differences between all measured parameters. However, the author concludes that their study suggests a preliminary conclusion that coordinated care in the form of the analyzed model has a positive impact on the outcomes of patients with HTN and comorbidities.

7.      The repetition of the results in the tables and the text should be removed.  

8.      What about other demographic data including smoking and alcohol? How the authors control these factors?

Round 2

Reviewer 1 Report

Comments and Suggestions for Authors

Thank you for addressing the comments

Author Response

Thank you for your support and valuable comments.

Reviewer 2 Report

Comments and Suggestions for Authors

The authors address my comments

Author Response

(The authors gave the same response as above.)
